

# Ether cleavage and chemical removal of SU-8

**José M. Ripalda⋆, Raquel Álvaro, and María Luisa Dotor**

Instituto de Micro y Nanotecnología, IMN-CNM, CSIC (CEI UAM+CSIC),
Isaac Newton 8, E-28760, Tres Cantos, Madrid, Spain

⋆ j.ripalda@csic.es

## Abstract

The high chemical stability of SU-8 makes it irreplaceable for a wide range of applications, most notably as a lithography photoresist for micro and nanotechnology. This advantage becomes a problem when there is a need to remove SU-8 from the fabricated devices. Researchers have been struggling for two decades with this problem, and although a number of partial solutions have been found, this difficulty has limited the applications of SU-8. Here we demonstrate a fast, reproducible, and comparatively gentle method to chemically remove SU-8 photoresist. An ether cleavage mechanism for the observed reaction is proposed, and the hypothesis is tested with *ab initio* quantum chemical calculations. We also describe a complementary removal method based on atomic hydrogen inductively coupled plasma.

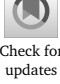

# 1 Introduction

## 1.1 Motivation

SU-8 is an epoxy based chemically amplified negative photoresist of widespread commercial and academic interest [1–6]. As a photoresist, SU-8 combines high chemical stability, high resolution, and high aspect ratio. Our motivation for studying SU-8 is its possible use as a sacrificial mold for the electroplating of silver contacts on solar cells and other optoelectronic devices. Silver is the metal with the highest electrical conductivity. Furthermore, integrated over the solar spectrum, silver has the highest reflectivity and lowest parasitic absorption, allowing for top contacts designs with lower optical losses. The electroplating of high density silver with optimal electrical and optical properties requires strongly basic cyanide electrolytes that instantly dissolve most photoresists, except SU-8, owing to its exceptional chemical stability, a property that becomes a problem after electroplating, as SU-8 would hinder further processing steps required for device fabrication.

## 1.2 State of the art for SU-8 removal

One of the most recent proposals for removal of SU-8 is laser ablation followed by an oxidizing plasma [7]. Han *et al.* adopted this method after finding unsatisfactory results from burning and ashing at temperatures ranging from 500°C to 1000°C. Most SU-8 removal methods rely on highly oxidizing conditions. Unfortunately, these conditions would cause damage on a wide range of materials of technological interest, limiting the applicability of the technique. Examples of these methods are various types of burning, oxidizing plasmas [7,8], piranha etching [9], and ozone exposure [10]. Certain molten salts have also been found to completely remove SU-8 at temperatures above 300°C, unfortunately, these extreme conditions are also incompatible with a wide range of applications [11]. Other proposed methods are mostly mechanical, such as water jetting [12], bead blasting [9], $CO_2$ at extreme conditions [13], or plasma and ion beam sputtering. Organic solvents can sometimes lift off the photoresist from the substrate, at least partially [14]. This process is more effective when there is poor adhesion with the substrate, little resist cross linking, and specially if usage of a sacrificial layer is possible. A disadvantage of all the mechanical methods mentioned above is that the photoresist removal is often incomplete, inhomogeneous, and irreproducible, as it depends on the geometry of the resist features among many other factors. Thus a chemical method to quickly and completely dissolve the cross-linked resist would be highly desirable. Piranha etching and oxygen plasmas meet this requirement but are highly oxidizing and thus incompatible with a wide range of materials. Without prior treatment, a 5 hour immersion in fuming sulfuric acid of nickel parts electroplated through an SU-8 mask results in cracking of the photoresist into small pieces, resulting in at least partial removal after rinsing with solvents [15]. Unfortunately, in this case the resist is cracked rather than dissolved, and thus removal is often incomplete. Here we show that if the SU-8 is pretreated with a water based dimethylamine solution, the resist immediately dissolves after subsequent immersion in sulfuric acid, leaving no residues and allowing for very short exposures to the acid.

## 1.3 SU-8 chemistry

As illustrated in Fig. 1, SU-8 polymerization is activated by epoxide ring opening in the presence of protons (acids) [16]. When used as a photoresist, SU-8 is mixed with up to 10% Triarylsulfonium hexafluoroantimonate salt as a photoacid generator [3,17]. Ring opening after protonation of the epoxide groups results in a carbocation [18]. Polymerization is due to the subsequent reaction of the carbocation with other epoxide groups [16]. Each carbocation

can participate in polymerization reactions with multiple epoxide groups. This is what makes SU-8 a chemically amplified photoresist, as a single photon generating a single proton leads to multiple polymerization events [18]. Each SU-8 monomer has 8 epoxide functional groups and thus can be linked with up to 8 other SU-8 monomers, yielding a highly cross-linked and dense structure unlike that of most linear polymers. The weakest link in polymerized SU-8 are the C-O-C single (ether) bonds between SU-8 units resulting from epoxide ring opening and cross-linking. These are extremely inert bonds, only slightly weaker than the diamond-like (sp$^3$) and graphene-like (sp$^2$) C-C bonds in the rest of the SU-8 structure [19].

### 1.4 Ether bond cleavage

The cleavage of ether bonds has been [20], and still is, an important area of research in organic chemistry [21]. Most current interest on ether cleavage stems from efforts towards valorization of wood and organic waste [22]. Methods for ether cleavage include acid cleavage, alkali metal cleavage, nucleophilic cleavage, heterogeneous and homogeneous catalytic cleavage [23–25], electrochemical cleavage [26, 27], and photochemical cleavage. Unfortunately, most procotols for ether cleavage are specific for small molecules with certain aryl or alkyl ethers including specific functional groups. It seems difficult to foresee how applicable these ether cleavage methods are in the case of highly polymerized SU-8, mostly due to the steric hindrance effects, as the dense polymer structure protects the sites of attack. After an extensive review of the current literature we have not found any successful application of these chemical ether cleavage methods for SU-8 removal. The method here reported was discovered accidentally after unsuccessfully attempting to lift off the resist with dimethylamine as a solvent, and subsequently immersing the sample in sulfuric acid.

## 2 Methodology

### 2.1 Sample preparation

The SU-8 removal procedure here described was found accidentally during development of a new GaAs solar cell fabrication process. Although only some aspects of the process are relevant to the results here presented, a brief description of the solar cell fabrication process is given here to put our results in context. To reduce surface oxidation, the samples were transferred in a dry nitrogen atmosphere to a vacuum electron beam evaporation chamber for seed layer metalization after semiconductor layer deposition in a molecular beam epitaxy reactor. Contamination and oxidation of the metal semiconductor interface can result in degradation of the contact resistance. These metal layers also serve as a seed layer for subsequent silver electroplating. Silicon (001) substrates were also used in experiments to test the removal speed for different SU-8 formulations. Both types of substrates were metallized with a 60 nm Ti adhesion layer and a 5 nm Pt top layer. To improve adhesion of the SU-8 photoresist to the metallized substrate, the samples were treated by immersion in a solution of 57.7 mg of 16-Mercaptohexadecanoic acid in 385 ml of ethanol, with 10 ml of deionized water, and 5 ml of acetic acid at 70°C for 10 min. Ultrasonication at 60°C for 5 min was repeated before each use of the solution to avoid the presence of agglomerates. Without this treatment partial delamination was often observed between the SU-8 photoresist and the metal surface after development of the lithography pattern. This hydrophilyc alkyl-thiol was chosen due to the well known tendency of alkyl thiols to form self assembled monolayers on noble metal surfaces. Prior to SU-8 spin coating, the samples were rinsed by spraying with isopropyl alcohol while spinning at 4000 rpm and then blow dried with nitrogen.

    The photovoltaic devices additionally had a pattern of silver wires electrodeposited through

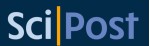

the lithographically defined SU-8 mask. A silver cyanide based electrolyte was used for electroplating. In our tests this electrolyte was found to be incompatible with a number of photoresists, but no effect was apparent on SU-8. Removal of SU-8 from samples not exposed to the electrolyte was also successful and no significant effect was found in the removal speed regardless of exposure to the electrolyte. After SU-8 removal, the Ti seed layer is removed with a quick dip in diluted HF acid and the GaAs contact layer is removed with a citric acid and hydrogen peroxide etchant.

Figure 1: Upon UV light exposure, protons released by a photoacid react with one of the 8 epoxide functional groups in the SU-8 monomer unit [16]. Protonation breaks the epoxide cycle, resulting in a carbocation and a hydroxy group at the end of the carbon chain. The carbocation subsequently reacts with another epoxide leading to polymerization, and persists after polymerization [18]. SU-8 is thus a chemically amplified photoresist, as a single photon can trigger several polymerization events. According to our DFT calculations, the more energetically favourable position for the carbocation is immediately adjacent to the ether group.

## 2.2 SU-8 photoresist UV exposure

A correct ultraviolet light (UV) exposure is critical not only for successful photolithographic pattern definition but also for subsequent SU-8 chemical removal. Gaudet *et al.* have determined the critical exposure dose threshold for polymerization to be 49.4 mJ cm$^{-2}$ [28]. Because the ultraviolet radiation is strongly attenuated due to absorption in the SU-8 photoresist, the exposure dose needs to be adjusted for thickness to ensure that buried layers are exposed above the threshold for complete polymerization. This can be done using the absorption coefficient at 365 nm wavelength determined by Gaudet *et al.*, which initially is 38 cm$^{-1}$ and increases up to 49 cm$^{-1}$ after very high exposures (415 mJ cm$^{-2}$). As an approximation, we have assumed the absorption coefficient to be 43 cm$^{-1}$. It is important to emphasize here the need to filter short wavelength radiation ($< 365$ nm) when using non-monochromatic light sources, as severely overexposed surface layers are obtained otherwise. This leads to fabrication defects and more difficult SU-8 removal. Interference standing waves due to the light reflection in the substrate can lead to inhomogenous photoresist exposure, requiring a higher UV dose and leading to difficulties in SU-8 removal. UV reflection was 45% for the chosen Pt/Ti metallization in our experiments. We have tested the here described chemical removal protocol on three different SU-8 formulations by MicroChem - Kayaku Advanced Materials: SU-8 2002, SU-8 2005, and SU-8 2150. SU-8 was spin coated at 4000 rpm for 40 s with a 10 s ramp up time. A soft pre-bake of the coating was done by ramping the temperature up for 10 min, sustaining 95°C for 2 min (30 min for thickness $> 100$ $\mu$m), and then ramping down for 20 min. After UV exposure at a dose adequate for the desired film thickness, a post exposure bake was done at 70°C for 2 min, then 95°C for another 2 min (12 min for thickness $> 100$ $\mu$m) and then ramping down for 20 min. The conditions used for SU-8 polmerization are summarized in Table 1.

## 2.3 SU-8 chemical removal procedure

Our experimental protocol for SU-8 removal starts with a 10 min immersion in a 40% solution of dimethylamine in water at 70°C. Higher temperatures are not recommended as the highly volatile dimethylamine solute lowers the boiling point of the solution, and vigorous boiling leads to rapid loss of the dimethylamine. Then we proceed by rinsing in deionized water, blow drying in N$_2$, and immersion in concentrated sulfuric acid followed again by rinsing in deionized water and blow drying in N$_2$. Depending on thickness, UV exposure and post-exposure bake conditions, the immersion time required for complete removal ranges from 5 seconds to 30 minutes.

## 2.4 Hydrogen inductively coupled plasma removal procedure

As an alternate procedure for SU-8 removal we describe here the procedure and resulting attack rate for SU-8 removal with a hydrogen inductively coupled plasma (ICP). This attack is useful for thin SU-8 layers (5 $\mu$m or less) or to clean up residues left after chemical removal of thick and highly polymerized SU-8 layers. Tanaka *et al.* recently attempted SU-8 removal with a hot-wire atomic hydrogen source, but resulted in incomplete removal even at elevated temperatures [29]. Here we demonstrate complete removal at ambient temperature using a hydrogen inductively coupled plasma. Because of the low sputtering yield of hydrogen, the removal mechanism is not expected to be due to impact displacement, as is the case with argon plasmas, but due to the chemical reaction of plasma activated hydrogen. Consequently very low damage can be expected in delicate semiconductor substrates such as GaAs and InP. The fraction of the intial projectile kinetic energy transferred to the target is given by the masses of the projectile ($m_H$) and the target ($m_{Ga}$) as $4m_H^2/(m_{Ga} - m_H)^2 = 8 \times 10^{-4}$. As the mean

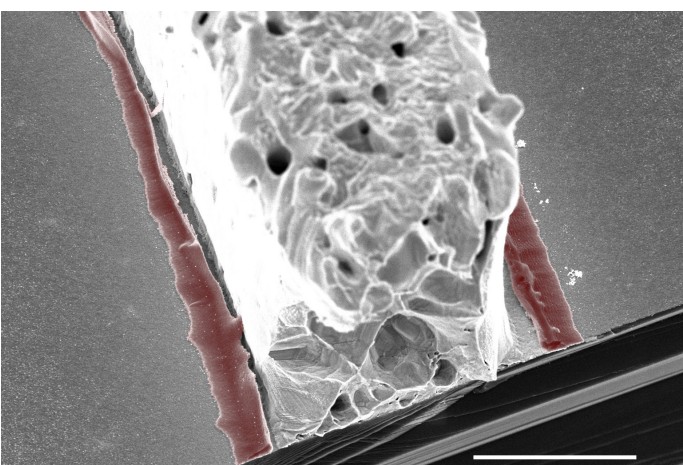

Figure 2: Scanning electron microscopy image of a silver wire electrodeposited on a GaAs photovoltaic device through an SU-8 photoresist mask that was then removed by an ICP RIE hydrogen plasma. SU-8 residues (colored red) can be observed by the silver wires as the ICP RIE attack rate is also a function of the surface topography. The white scale bar is 2 $\mu$m long.

ion energy in low pressure hydrogen plasmas is typically well below 100 eV [30], hydrogen ions typically transfer less than 0.1 eV kinetic energy to the target atom, and the effects of the hydrogen ICP plasma are mostly chemical due to protonation.

Hydrogen plasma removal of SU-8 was done in an Oxford Instruments PlasmaLab-80 Plus reactive ion etch machine (RIE) with 300 W ICP power and 100 W radiofrecuency power on a chilled sample plate at 0°C with a hydrogen flow rate of 30 SCCM. Before sample introduction the vacuum chamber was conditioned by igniting the plasma with the addition of 5 SCCM of nitrogen, but only pure hydrogen was used after the sample was introduced in the vacuum chamber. The plasma was ignited at 90 mbar and then the pressure was lowered to 30 mbar. A sample polarization (DC bias) of 264 V was measured. The resulting attack rate was 2 nm/s. Although this attack rate is enough for thin SU-8 layers, Fig. 2 illustrates the need for a wet chemical reaction for removal, as some parts of the sample geometry make it difficult remove the polymer with hydrogen ICP RIE. In this case, silver wires that have been electrodeposited using the SU-8 photoresist as a mask are shadowing some parts of the polymer, slowing the ICP RIE attack rate in certain areas. As shown below, the hydrogen ICP RIE technique is also useful to remove surface residues left after wet chemical attack of thick and highly polymerized SU-8 layers.

## 2.5 DFT methodology

Density Functional Theory (DFT) based quantum chemical calculations were performed with Gaussian 16 [31]. The hybrid meta generalized gradient approximation functional M06-2X [32] was used for all calculations. A triple zeta 6-311++G(d,p) basis set was used for single point energies after geometry optimization and frequencies were obtained with the 6-31G(d) double zeta basis set. The conductor-like polarizable continuum model was used as the self-consistent reaction field [33]. This methodology was found by Sturgeon *et al.* to yield excellent agreement with the Complete Basis Set CBS-QB3 benchmark method for model molecules of interest for ether cleavage [22]. Gaussian input and output files are publicly available at https://zenodo.org/record/6351619.

Table 1: SU-8 removal time as a function of formulation and photoresist thickness. The critical dose is calculated as prescribed by Gaudet *et al.* [28]

| Formulation | Thickness | Used Dose | Critical Dose | Postbake time | Removal time |
|---|---|---|---|---|---|
| SU-8 2002 | 1.8 $\mu$m | 108.0 mJ/cm$^2$ | 49.4 mJ/cm$^2$ | 2 min | < 1 min |
| SU-8 2005 | 3.8 $\mu$m | 112.5 mJ/cm$^2$ | 49.8 mJ/cm$^2$ | 2 min | 1 min |
| SU-8 2150 | 118 $\mu$m | 135.0 mJ/cm$^2$ | 81.4 mJ/cm$^2$ | 12 min | 27 min |

## 3 Results

### 3.1 SU-8 chemical removal results

In case of overexposure, the time for SU-8 removal greatly increases, and conversely in case of subexposed photoresist the removal time is shorter than the removal time required for a well exposed photoresist. To obtain a properly polymerized photoresist there is an optimal ultraviolet exposure dose and resist bake temperature for each photoresist thickness. If all the experimental parameters are independently scanned, then in most cases the results obtained correspond to sub-optimal process conditions that are not of interest. We have thus selected optimal processing conditions for 3 different SU-8 photoresist film thicknesses and tested our removal process under these conditions. We find that the sulfuric acid immersion time required for SU-8 removal increases with photoresist thickness due to the required higher UV exposure time and longer post exposure bake time (Table 1).

The depolymerization reaction is immediately apparent when SU-8 samples that have been treated with dimethylamine are immersed in sulfuric acid (Fig. 3). In our tests with SU-8 thicknesses ranging from 1 $\mu$m to 5 $\mu$m, a 1 min immersion suffices to completely remove SU-8 in all cases and no etching has been observed for any of the other materials in our samples, including Ag, Pt, Ti, Si, and GaAs (see Fig. 4). A wide range of metals and dielectrics can withstand a short immersion in concentrated sulfuric acid (aluminum and chromium being notable exceptions). Although many metals, dielectrics and semiconductors are unaffected by exposure to concentrated sulfuric acid, some semiconductor surface/interface oxides and metal tarnish layers are readily etched, and this can be of consequence for device processing. No residues are found after removal of thin (< 5 $\mu$m) SU-8 layers. Compositional contrast using back-scattered electron detection shows only two components after SU-8 removal, the electrodeposited silver and the Pt/Ti seed layer (see Fig. 5). Any organic residue would be observed as a dark stain on the Pt/Ti seed layer. What appears to be a small (200 nm) particle of organic matter can be observed near the center of the top edge of Fig. 5c. The circular protrusions in the seed layer do not show any compositional contrast and appear to be bubbles or bumps under the thin (65 nm) Pt/Ti seed layer.

For UV exposure doses slightly above the critical value determined by Gaudet *et al.* [28], the SU-8 removal is nearly instantaneous, but longer acid immersion times are required for higher UV exposure doses. In the case of thick and overexposed layers, ultrasonic agitation, longer acid submersion times (30 min) and moderate heating (40°C) are required. In the later case, discontinuous residues were observed on the surface by optical microscopy after SU-8 removal (Fig. 6). Profilometer measurements indicate a 400 nm height for the observed dendrites. Energy dispersive spectra of the X-rays emitted under examination with a scanning electron microscope indicate no trace of the antimony based salt used as a photoacid in SU-8. These residues are eliminated without difficulty after 480 s of attack in the RIE ICP hydrogen plasma described in Section 2.4, suggesting the residues are organic matter.

The effect of a number of other organic solvents, such as tetrahydrofuran, gamma-butyrolactone, and dimethylformamide, was also tested, and none of them was found to permanently alter the reactivity of SU-8, as was the case with dimethylamine.

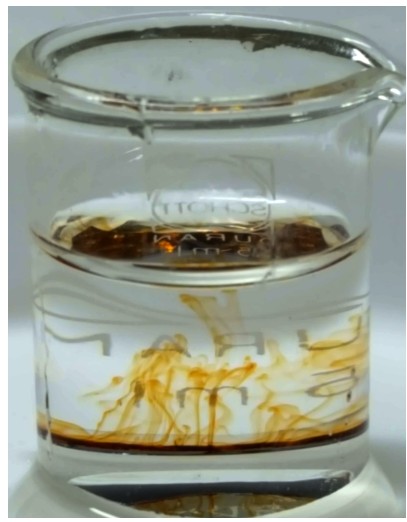

Figure 3: After treatmentment with a water based dimethylamine solution, the SU-8 resist immediately dissolves after subsequent immersion in sulfuric acid, allowing for very short exposures to the acid. The reaction products float to the top of the solution.

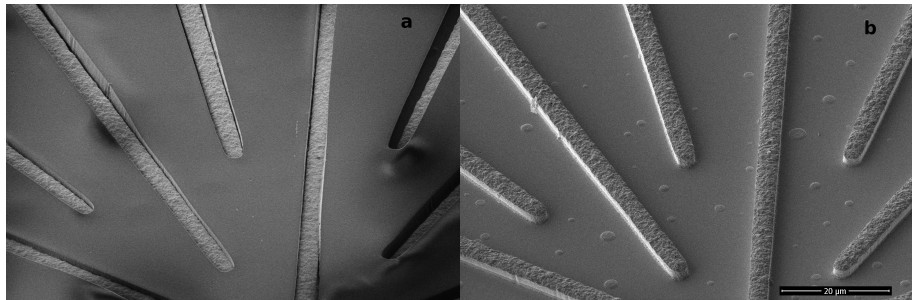

Figure 4: Scanning electron microscopy image of the same spot on a solar cell contact electrodeposited using an SU-8 photolithographic mask (a), with SU-8 later removed by exposure to dimethylamine and subsequent immersion in sulfuric acid (b).

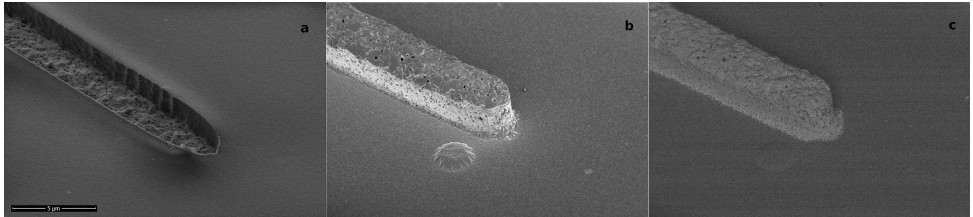

Figure 5: Scanning electron microscopy image of the same spot on a solar cell contact electrodeposited using an SU-8 photolithographic mask (a), with SU-8 later removed by exposure to dimethylamine and subsequent immersion in sulfuric acid (b). Compositional contrast using back-scattered electron detection shows only two components after SU-8 removal, the electro-deposited silver and the Pt/Ti seed layer (c).

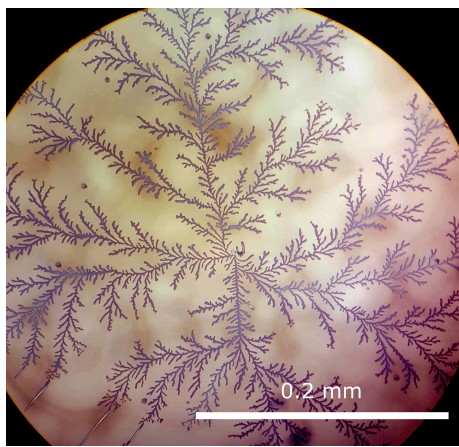

Figure 6: In thick ($> 100$ $\mu$m) and highly polymerized samples, discontinuous residues were observed on the surface by optical microscopy after SU-8 removal. No residues were detected after removal of thinner SU-8 layers. These residues were determined to have a thickness of 400 nm by profilometer measurements, and were removed without difficulty by RIE ICP hydrogen plasma. The diameter of the field of vision is 400 $\mu$m.

## 3.2 Hypothesis on reaction mechanism and DFT results

Sulfuric acid is known to catalyze ether cleavage, leaving both sides of the split molecule terminated with OH groups [22]. All the carbon atoms participating in the ether bonds resulting from SU-8 polymerization are sp$^3$ hybridized. This is relevant because sp$^2$ hybridized carbon atoms participating in aromatic rings are partially protective against acid induced ether cleavage, so SU-8 can *a priori* be expected to be susceptible to depolymerization in sulfuric acid. But exposure of SU-8 to sulfuric does not result in depolymerization and dissolution of the polymer, much to the contrary SU-8 hardens, blackens, and cracks, making a complete clean up even more difficult. A drastic change in the reactivity of SU-8 in sulfuric is observed after exposure to dimethylamine.

While it is difficult to determine the exact role played by dimethylamine, the initial rationale for using this chemical was the well known tendency of amines to link with epoxide functional groups (Fig. 7) [34–36]. While molecules with multiple exposed N-H bonds are often commercially used for inducing cross linking of monomers with epoxide functional groups, it was foreseen that a small molecule with a single available N-H bond would terminate the polymer chain avoiding further polymerization during subsequent processing. Dimethylamine is the smallest molecule meeting this requirement. So dimethylamine can be expected to be protective against further polymerization when the SU-8 is exposed to sulfuric. The highly reactive carbocations resulting from epoxide polymerization can also be expected to be a reaction site for dimethylamine. Such reaction is strongly favoured by a large reduction in the Gibbs free energy as shown in Fig. 8. The drastic and permanent change induced by exposure to dimethylamine in the depolymerization rate of SU-8 when reacting with sulfuric acid suggests that dimethylamine is also directly involved at the ether cleavage reaction site. The weaker links in the polymer backbone are the ether bonds immediately adjacent to a carbocation and in the vicinity of an hydroxyl group as shown in Fig. 1. Furthermore, the fact that previous exposure of SU-8 to dimethylamine suffices to permanently change the reactivity of SU-8, led us to hypothesise that dimethylamine reacts with the carbocations in the SU-8 backbone, facilitating the ether cleavage in sulfuric acid. This hypothesis has been tested by quantum mechanical calculations of the energetics of the reaction pathway. It is important to under-

Figure 7: In our SU-8 removal protocol, dimethylamine reacts with the remaining epoxide groups [34–36], inhibiting the further polymerization that can otherwise be expected under acidic conditions.

score the limitations this type of calculations: the results, summarized in Fig. 8, are consistent with the proposed hypothesis, but this does not confirm the hypothesis as correct, it merely confirms the hypothesis as energetically favourable, as only a few of all the possible reaction pathways can be explored, and no information is obtained on reaction kinetics. The cleavage of ether linkages in sulfuric acid has been studied by Sturgeon *et al.* with quantum mechanical calculations and experimental measurements of reaction kinetics of model molecules. All of the ether bond cleavage reaction pathways studied by Sturgeon *et al.* start with protonation of an hydroxyl group in the vicinity of the target ether bond. In our case, protonation of the hydroxyl group is hindered by Coulomb repulsion due to the presence of the carbocation in the immediate vicinity of the hydroxyl group and the target ether bond, with a free energy cost of 3.77 kcal/mol as deduced from our DFT calculations. Our calculations also reveal that the attachment of dimethylamine at the carbocation site is energetically very favourable (-29.87 kcal/mol), and results in a threefold reduction of the energy cost of the required reaction intermediate for hydroxyl group protonation (0.92 kcal/mol vs. 3.77 kcal/mol). Hydroxyl protonation after dimethylamine attachment leads to an unstable intermediate product with one positive charge at each side of the ether bond, and the electrostatic repulsion between these two charges spontaneously rips apart the ether bond with a free energy change of -13.72 kcal/mol.

Figure 8: Changes in the Gibbs free energy as SU-8 is protonated by the acid. At the top and to the right is shown the intermediate reaction product required for ether cleavage without previous exposure to dimethylamine. At the bottom and to the right is shown the reaction intermediate for ether cleavage after exposure to dimethylamine.

## 4    Conclusion

We achieve chemical removal of SU-8 photoresist without damaging the fabricated devices with a comparatively gentle method based on sequential exposure to dimethylamine and concentrated sulfuric acid. Drawing inspiration from the ample literature on ether cleavage for the revalorization of lignin from wood, we propose a hypothesis for the reaction mechanism and test its validity with *ab initio* quantum chemical calculations. Perhaps further work on this line of research can give back to the field of ether cleavage for lignin processing by providing new methods or insights. But this work will primarily be of interest to technologists developing microelectronic and microelectromechanical devices.

## Acknowledgements

**Author contributions**    JMR conceived the research and wrote the first version of the manuscript. All authors contributed to the manuscript and to the experimental work.

**Funding information**    Funding was provided by the Comunidad de Madrid (P2018/EMT-4308), the Spanish Ministerio de Ciencia e Innovación (MCIU-AEI) and the European Union (FEDER) through projects PID2021-124193OB-C22 and TED2021-130623B-I00. We acknowledge the service from the MiNa Laboratory at IMN made possible by funding from CM (project S2018/NMT-4291 TEC2SPACE), MINECO (project CSIC13-4E-1794) and EU (FEDER, FSE).

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
