# Peer review of "Ether cleavage and chemical removal of SU-8"

_SciPost Chemistry, doi:SciPost Chem. 2, 001 (2023)_

## Round 1 · Referee Report · Anonymous (Referee 1) · 2022-6-5

Strengths

1- The topic is of clear interest and relevance to applications.

2- The introduction provides a clear and concise presentation of the knowledge of the field.

Weaknesses

1- The actual results presented are relatively scarce, and do not go in depth into any of the areas studied.

2- The claims made are overall too general and overreaching compared to the data obtained. For example, in the conclusion: "Three current technological issues with SU-8 have been addressed in this work" seems a very big claim compared to the results available.

3- The authors present a lot of hypotheses and suppositions, and often lack actual data to confirm or infirm them. As such, the manuscript is mostly speculative/argumentative, rather than scientific findings.

4- The DFT section, done at a reasonable but very routine level of theory, presents only one possible pathway. It does not compare pathways, or explain some of the choices in the pathway studied.

Report

The content of the paper is overall interesting, but it feels a bit like a mixed pot of very different results (surface preparation, UV exposure, removal, computational results), where none of the areas involved have really been pursued in depth: it does not mean that the findings have no merit, but it raises more questions than it answers.

Requested changes

1- The authors present, in the "Polymer-metal adhesion treatment" section, a "an in house developed procedure to chemically modify the metal surface […] described for the first time". However, this procedure is not explicitly compared to the other state-of-the-art methods (in the Methods section), and its impact on the sample obtained is not presented in the manuscript. Is the adhesion obtained better? (one supposes so) If so, how was this improvement characterized, what quantitative and qualitative factors were compared? Etc.

2- In the section about chemical removal, only three experimental conclusions are investigated and reported. It is not entirely clear to me why these three were chosen, and why the different factors were not investigated separately. It seems to be difficult to reach conclusions, as the authors do, on the basis of only these three experiments.

3- In that same section, how is "complete removal" characterized? Is it visual inspection, or are tests actually performed to quantify the removal? Figure 3 is not very useful in this regard, as a very macroscopic view of dissolution, with no real information provided.

4- Figure 4 needs a scale bar, the diameter of the field of vision is not sufficient for accurate measurements.

5- Section 3.2 and Figure 5: the authors present this reaction in the text as "a hypothesis", but the Figure is less clear and presents it as a fact. The author should perform analysis to confirm whether the tertiary amine is indeed observed after treatment with dimethylamine, which would either confirm the presence of this species (and validate the findings), or invalidate the hypothesis. More than half a page is spent presenting this hypothesis, so it would need to be checked experimentally.

6- This is a major point, which I do not understand: the mechanism proposed and studied by DFT implies the pre-existence of a stable carbocation, that exists as a stable species? The authors state in the caption of Figure 1 "The carbocation persists after polymerization", but that seems surprising to me, so it needs to be argue: how has it been observed and confirmed? what is the anion associated? This is a very unexpected hypothesis, so it needs a strong confirmation in my view.

7- In the discussion of the DFT section, the authors discuss a lowering of the "energy barrier", but transition states have not been characterized or discussed at all. The transitions states need to be determined, and the alternative pathways characterized as well, for comparison and confirmation that the chosen mechanism is representative. The current presentation is way too speculative.

7bis- The authors acknowledge so, saying: "the results are consistent with the proposed hypothesis, but this does not confirm the hypothesis as valid, it merely serves to gain confidence in its validity" and "only a few of all the possible reaction pathways can be explored, and only thermodynamic results on stability are easily achievable, results on reaction kinetics being much more challenging to obtain". But this is not true, there are a lot of DFT studies of chemical reactivity and mechanisms that study both intermediates and transitions states, and compare different mechanisms to conclude: this is, in fact, what DFT studies are routinely used for, in order to reach conclusions about reactivity. Just confirming one set of relative stabilities seems well under the typically bar for mechanistic studies, in this respect.

  • validity: low
  • significance: low
  • originality: ok
  • clarity: good
  • formatting: excellent
  • grammar: excellent

Author:  José M. Ripalda  on 2022-06-09  [id 2571]

(in reply to Report 1 on 2022-06-05)

We agree with all the comments and assessments made by the reviewer. Changes to the manuscript will be done when given the opportunity by the editors.

In the following we provide answers to each of the questions and suggestions raised by the reviewer.

  • "Three current technological issues with SU-8 have been addressed in this work" seems a very big claim compared to the results available.

Agreed. We will withdraw this statement.

  • The authors present a lot of hypotheses and suppositions, and often lack actual data to confirm or infirm them. As such, the manuscript is mostly speculative/argumentative, rather than scientific findings. The DFT section, done at a reasonable but very routine level of theory, presents only one possible pathway. It does not compare pathways, or explain some of the choices in the pathway studied.

We compare two pathways, with and without dimethylamine. We are mostly experimentalists. Our capabilities for DFT are limited, thus it would be imprudent for us to pursue a study beyond a very routine level of theory. Nevertheless we felt that the possible mechanism of the reaction required at least some discussion and thus we have included our modest DFT effort to support such discussion.

The mechanistic study is admittedly incomplete, but nevertheless we think it would be beneficial to the community to have these results published even if the mechanism of the discovered reaction has not been completely elucidated.

We have experimentally found that dimethylamine exposure induces a change in SU8 that makes it soluble in sulfuric acid, but definite proof of what is the mechanism of action is a matter for future research.

1- The authors present, in the "Polymer-metal adhesion treatment" section, a "an in house developed procedure to chemically modify the metal surface […] described for the first time". However, this procedure is not explicitly compared to the other state-of-the-art methods (in the Methods section), and its impact on the sample obtained is not presented in the manuscript. Is the adhesion obtained better? (one supposes so) If so, how was this improvement characterized, what quantitative and qualitative factors were compared? Etc.

We agree with the referee that our claims in regard to this aspect of our work would require more quantitative data, and thus we will withdraw most of our statements in regards to this aspect of our work. The adhesion improvement manifested itself by fewer delamination events at the SU8 / Pt interface. Before using the described process based on mercaptohexadecanoic acid self assembled monolayers we were using a process based on a commercial solution (Jentner Oxiprotect) that in some cases led to a complete lack of wetting during spin coating, particularly when attempting to coat with Brewer-Science bottom antireflective coating ARC I-con.

2 - In the section about chemical removal, only three experimental conclusions are investigated and reported. It is not entirely clear to me why these three were chosen, and why the different factors were not investigated separately. It seems to be difficult to reach conclusions, as the authors do, on the basis of only these three experiments.

We will include a discussion of the interdependence of the various experimental factors. The parameter window for high quality lithography is quite narrow and the combination of optimal parameters needs to be determined on a case by case basis. A large number of experiments were done to determine this combination of parameters for each photoresist thickness. A complete description of all experiments would only add confusion to our manuscript as many of those experiments correspond to suboptimal photoresist exposure conditions in a trial and error effort. We have chosen to report instead the experiments resulting in high quality photolithography and subsequent photoresist removal. In case of overexposure the time for removal greatly increases, and conversely in case of subexposed photoresist the removal time is not representative of the optimal process. If the experimental parameters are independently scanned, then in most cases the results obtained correspond to sub-optimal process conditions that are not of interest. As observed by the reviewer, this merits a more detailed discussion in the manuscript.

3- In that same section, how is "complete removal" characterized? Is it visual inspection, or are tests actually performed to quantify the removal? Figure 3 is not very useful in this regard, as a very macroscopic view of dissolution, with no real information provided.

Samples were examined after removal by optical microscopy, electrical contact resistance and in some cases with scanning electron microscopy and energy dispersive x-ray emission spectroscopy. A very thin residual layer (50 nm) cannot be ruled out as it can be easily pierced by the electrical contact probes, but a photoresist layer of more than 50 nm is easily detected as it changes the reflectivity of the underlying metal due to interference. So we will modify the manuscript to reflect that we experimentally verified in that in all cases a residue of no more than 2.5% of the initial thickness remains on the surface. The phrase "complete removal" will be withdrawn.

4- Figure 4 needs a scale bar, the diameter of the field of vision is not sufficient for accurate measurements.

A scale bar will be included.

5- Section 3.2 and Figure 5: the authors present this reaction in the text as "a hypothesis", but the Figure is less clear and presents it as a fact.

The reaction of dimethylamine with epoxide functional groups has repeatedly been verified in the literature. We will include sufficient references to support this claim. E.g.:

Epoxide Reactions, Thomas Bertolini, Journal of Chemical Education • Vol. 79 No. 7 July 2002

Highly Chemoselective Addition of Amines to Epoxides in Water Najmodin Azizi and Mohammad R. Saidi Org. Lett. 2005, 7, 17, 3649–3651 https://doi.org/10.1021/ol051220q

Assessing the Potential for the Reactions of Epoxides with Amines on Secondary Organic Aerosol Particles Santino J. Stropoli and Matthew J. Elrod, J. Phys. Chem. A 2015, 119, 40, 10181–10189 https://doi.org/10.1021/acs.jpca.5b07852

6- This is a major point, which I do not understand: the mechanism proposed and studied by DFT implies the pre-existence of a stable carbocation, that exists as a stable species? The authors state in the caption of Figure 1 "The carbocation persists after polymerization", but that seems surprising to me, so it needs to be argue: how has it been observed and confirmed? what is the anion associated?

This also merits further discussion in the manuscript with appropriate references to the literature. The carbocation results from the protonation of the epoxide functional group, this is a well known reaction of epoxide groups. It has also been widely reported that this carbocation subsequently reacts with other epoxide functional groups producing further polymerization, and it is the reason for SU8 being, by design, a chemically amplified photoresist, as a single photon results in the generation of a carbocation by the Triarylsulfonium/hexafluoroantimonate photoacid, and this single carbocation can lead to several polymerization events as it persists after polymerization. The anion associated to the carbocation is the SbF6- anion from the photoacid in the SU8 formulation. Appropriate references describing the role of carbocations in SU8 polymerization will be included.

Triarylsulfonium hexafluorophosphate salts as photoactivated acidic catalysts for ring-opening polymerisation January 2013Chemical Communications 49(12) DOI:10.1039/c2cc38114a

7- In the discussion of the DFT section, the authors discuss a lowering of the "energy barrier", but transition states have not been characterized or discussed at all.

We base our work on the previous computational mechanistic investigation of Sturgeon et al (ref. 16) in a similar case of acid catalyzed cleavage of ether links. Sturgeon first performed a Monte Carlo Multiple minimum (MCMM) conformational search optimizing structures with the semi-empirical PM6 Hamiltonian. All low lying conformers within 15 kcal/mol of the global energy minimum were selected for DFT calculations. We take advantage of this previous effort to directly explore the most promising reaction pathways. But even in the case of the extensive exploration by Sturgeon et al., only stable configurations were studied as semiempirical methods are required (for cost reasons) for such an extensive exploration, and these are not adequate for transition states. The study of possible transition states is indeed possible and often done, but unless it can be shown that all possible transition states have been systematically studied, the effort is not justified. In practice this usually done in particularly simple cases. In complex cases such as the present one, an exhaustive search of all possible transition states is well beyond our capabilities. The presented DFT calculations do however show that the proposed pathways are energetically feasible.

We are grateful for the effort made by the reviewer and the useful feedback.

---

## Round 1 · Referee Report · Anonymous (Referee 2) · 2022-6-23

Strengths

1. The work is interesting and undoubtedly relevant for various communities working on nanotechnology, photovoltaics, electronics, etc…
2. The need for the present work is justified in the introduction based on the well-presented state of the art.
3. Different challenges of the optical lithography using SU-8 are targeted. The reported process is simple, based on widely available and unexpensive chemicals and is compatible with a wide range of materials, making it technologically relevant.

Weaknesses

1. Some claims in the text have little or no proof based on the reported results. Many statements need to be supported by references, experimental proof or simulations. More characterization would be desirable.
2. The scope of the paper is too wide to go deep into the details.

Report

The content of the paper is interesting and the results presented by the authors have a high technological relevance. The work meets the general acceptance criteria and presents a breakthrough on a previously-identified and long-standing research stumbling block. Therefore, I recomend it for publication in Scipost Chemistry provided that the questions below are clarified and/or corrected in the manuscript.

Requested changes

1. The main focus of the paper is the effectiveness of the chemical etching procedure discovered by the authors to remove SU-8 residues after lithography processes. The authors claim that their method is effective with the great advantage of being gentler for a wide range of materials and devices than previously reported procedures. However, no evidentiary result is provided in the manuscript to support this claim at all, apart from figure 3, which gives very little information. Characterization of the sample after processing should be therefore provided. At the very least, an SEM image after SU-8 dissolution should be given for comparison with figure 2. A good way to present it could be to move figure 2 to the results section and group it with the SEM picture of the final result. This is especially relevant for photovoltaic devices as the cells need to be further processed after finger deposition to add antireflection coatings and cleanness is crucial to achieve the desired optical properties.
2. The statements in section 1.3 (SU-8 chemistry and figure 1), which set the framework for the DFT investigation in section 3.2, are not obvious to me and should be supported by references.
3. A minor issue is that the sample preparation procedure in section 2.1 is not clear enough in my view. Are the samples metallized with Ti/Pt before the SU-8 deposition? Why? I would expect metallization to proceed after the deposition and development of the resist. Is that to test metal-polymer adhesion treatments? Is the top surface of the solar cells also metallized before photolithography? I encourage the authors to clarify these questions.
4. Section 2.2 (Polymer-metal adhesion treatment) looks a bit speculative. The authors should provide evidence of the formation of the self-assembled monolayer or support this assumption with references. Molecular self-assembly of alkyl thiols on metal surfaces is a widely studied matter so this minor issue should be easy to solve. More important is to clarify if there is evidence of the improvement of polymer-metal adhesion. Does the polymer delaminate during the baking in the absence of the self-assembled monolayer? Is the adhesion tested somehow or just proven by delamination (or not) during further processing? This part is interesting; however, it needs to be reformulated if the authors want to fit it properly in the scope of the present paper. Alternatively, as it is not the main issue addressed in the paper and not even discussed in the result section, the authors could further investigate and develop this part and leave it for another publication without detracting too much from the relevance of this paper.
5. As the authors explain in section 3.2, thermodynamical data from their DFT calculations does not proof that the proposed mechanism is the actual mechanism behind the observed results as transition states may play an important role and favor other reaction pathways. To my view and because of the technological relevance of the developed process, it is enough for the present work to prove that this is a feasible and likely mechanism with the performed calculations. However, the quality and significance of this work would be greatly enhanced if the authors provide further evidence supporting their hypothesis, either by more sophisticated simulations or by characterization of the dissolution with analytical chemistry methods.

---

## Round 2 · Referee Report · Anonymous (Referee 2) · 2022-12-7

Strengths

  1. The work is interesting and undoubtedly relevant for various communities working on nanotechnology, photovoltaics, electronics, etc…
  2. The need for the present work is justified in the introduction based on the well-presented state of the art.
  3. Different challenges of the optical lithography using SU-8 are targeted. The reported process is simple, based on widely available and unexpensive chemicals and is compatible with a wide range of materials, making it technologically relevant.

Report

The content of the paper is interesting, and the results presented by the authors have a high technological relevance. The work meets the general acceptance criteria and presents a breakthrough on a previously identified and long-standing research stumbling block. The authors answered all the questions and remarks in my first report; the manuscript is clearer, much less speculative and without irrelevant information. The paper has improved in quality during the peer-review process. Therefore, I congratulate the authors and recommend the present version of the article for publication in SciPost Chemistry.

---

## Round 2 · Author Response

We agree with all the comments and suggestions made by the reviewers and we have changed the manuscript accordingly.

In the following we provide answers to each of the questions and suggestions raised by the reviewers.

Reviewer 1: "Three current technological issues with SU-8 have been addressed in this work" seems a very big claim compared to the results available."

Agreed. We have withdrawn this statement.

"The DFT section, done at a reasonable but very routine level of theory, presents only one possible pathway. It does not compare pathways, or explain some of the choices in the pathway studied."

We are mostly experimentalists. Our capabilities for DFT are limited, thus it would be imprudent for us to pursue a study beyond a very routine level of theory. Nevertheless we felt that the possible mechanism of the reaction required at least some discussion and thus we have included our modest DFT effort to support such discussion. We compare two pathways, with and without dimethylamine. We think it would be beneficial to the community to have these results published even if the mechanism of the discovered reaction has not been completely elucidated. We have experimentally found that dimethylamine exposure induces a change in SU8 that makes it soluble in sulfuric acid, but definite proof of what is the mechanism of action is a matter for future research.

"The authors present, in the "Polymer-metal adhesion treatment" section, a "an in house developed procedure to chemically modify the metal surface […] described for the first time". However, this procedure is not explicitly compared to the other state-of-the-art methods (in the Methods section), and its impact on the sample obtained is not presented in the manuscript. Is the adhesion obtained better? (one supposes so) If so, how was this improvement characterised, what quantitative and qualitative factors were compared? Etc."

We agree with the referee that our claims in regard to this aspect of our work would require more quantitative data, and thus we have withdrawn the "Polymer-metal adhesion treatment" section. The adhesion improvement manifested itself by fewer delamination events at the SU8 / Pt interface.

"In the section about chemical removal, only three experimental conclusions are investigated and reported. It is not entirely clear to me why these three were chosen, and why the different factors were not investigated separately. It seems to be difficult to reach conclusions, as the authors do, on the basis of only these three experiments."

This aspect of our work had not been properly described and has now been revised including a discussion of the interdependence of the various experimental factors. SU-8 photoresists are commercially available at standard dilutions that correspond to certain photoresist thicknesses under optimal spin coating conditions. We have tested three of the most commonly used commercial solutions MicroChem SU-8 2002, 2005, and 2150. Under optimal spin coating conditions these correspond respectively to photoresist thicknesses of 2, 5, and 150 micron. For each photoresist thickness there is an UV optimal exposure dose and baking times and temperatures. The parameter window for high quality lithography is quite narrow and the combination of optimal parameters needs to be determined on a case by case basis by trial and error depending on the characteristics of the laboratory equipment and the requirements of the sample fabrication process. Doing SU-8 chemical removal experiments on the whole parameter space would lead to most experiments being done in conditions where the photoresist is severely over or under polymerised. This would have an impact on the SU-8 removal time, but would also make the photoresist unusable for device processing. Thus we only present results for the optimal process parameter combinations corresponding to the three SU-8 formulations that we had available in our laboratory.

"In that same section, how is "complete removal" characterized? Is it visual inspection, or are tests actually performed to quantify the removal? Figure 3 is not very useful in this regard, as a very macroscopic view of dissolution, with no real information provided."

Electro microscopy (SEM) images before and after SU-8 removal are now included in the manuscript. In most cases no residues can be detected with SEM microscopy in compositional contrast mode (backscattered electrons), but the phrase "complete removal" has been withdrawn.

"Figure 4 needs a scale bar, the diameter of the field of vision is not sufficient for accurate measurements."

A scale bar is now included.

"Section 3.2 and Figure 5: the authors present this reaction in the text as "a hypothesis", but the Figure is less clear and presents it as a fact."

The reaction of dimethylamine with epoxide functional groups has previously been verified in the literature. We now include references to support this claim.:

  • Epoxide Reactions, Thomas Bertolini, Journal of Chemical Education • Vol. 79 No. 7 July 2002

  • Highly Chemoselective Addition of Amines to Epoxides in Water Najmodin Azizi and Mohammad R. Saidi Org. Lett. 2005, 7, 17, 3649–3651 https://doi.org/10.1021/ol051220q

  • Assessing the Potential for the Reactions of Epoxides with Amines on Secondary Organic Aerosol Particles Santino J. Stropoli and Matthew J. Elrod, J. Phys. Chem. A 2015, 119, 40, 10181–10189 https://doi.org/10.1021/acs.jpca.5b07852

"This is a major point, which I do not understand: the mechanism proposed and studied by DFT implies the pre-existence of a stable carbocation, that exists as a stable species? The authors state in the caption of Figure 1 "The carbocation persists after polymerization", but that seems surprising to me, so it needs to be argue: how has it been observed and confirmed? what is the anion associated?"

The carbocation results from the protonation of the epoxide functional group, this has previously been verified in the literature. It has also been previously reported that this carbocation subsequently reacts with other epoxide functional groups producing further polymerization, and it is the reason for SU8 being, by design, a chemically amplified photoresist, as a single photon results in the generation of a carbocation by the Triarylsulfonium/hexafluoroantimonate photoacid, and this single carbocation can lead to several polymerization events as it persists after polymerization. The anion associated to the carbocation is the SbF6- anion from the photoacid in the SU8 formulation. References describing the role of carbocations in SU8 polymerization are now included.

  • Teh, W. & Duerig, U. & Drechsler, U. & Smith, Charles & Guentherodt, H.-J. (2005). Effect of low numerical-aperture femtosecond two-photon absorption on (SU-8) resist for ultrahigh-aspect-ratio microstereolithography. Journal of Applied Physics. 97. 10.1063/1.1856214.

  • Triarylsulfonium hexafluorophosphate salts as photoactivated acidic catalysts for ring-opening polymerisation January 2013Chemical Communications 49(12) DOI:10.1039/c2cc38114a

"In the discussion of the DFT section, the authors discuss a lowering of the "energy barrier", but transition states have not been characterized or discussed at all."

We base our work on the previous computational mechanistic investigation of Sturgeon et al (ref. 16) in a similar case of acid catalyzed cleavage of ether links. Sturgeon first performed a Monte Carlo Multiple minimum (MCMM) conformational search optimizing structures with the semi-empirical PM6 Hamiltonian. All low lying conformers within 15 kcal/mol of the global energy minimum were selected for DFT calculations. We take advantage of this previous effort to directly explore the most promising reaction pathways. But even in the case of the extensive exploration by Sturgeon et al., only stable configurations were studied as semiempirical methods are required (for cost reasons) for such an extensive exploration, and these methods are not adequate for transition states. The study of possible transition states is indeed possible and often done, but unless it can be shown that all possible transition states have been systematically studied, the effort is not justified. In practice this is usually done in particularly simple cases. In complex cases such as the present one, an exhaustive search of all possible transition states is well beyond our capabilities. The presented DFT calculations do however show that the proposed pathways are energetically feasible.

Reviewer 2:

"no evidentiary result is provided in the manuscript to support this claim at all, apart from figure 3, which gives very little information. Characterization of the sample after processing should be therefore provided. At the very least, an SEM image after SU-8 dissolution should be given for comparison with figure 2. A good way to present it could be to move figure 2 to the results section and group it with the SEM picture of the final result."

SEM images are now provided showing the same spot before and after SU-8 removal.

"The statements in section 1.3 (SU-8 chemistry and figure 1), which set the framework for the DFT investigation in section 3.2, are not obvious to me and should be supported by references."

References are now included to support the statements in section 1.3.

"A minor issue is that the sample preparation procedure in section 2.1 is not clear enough in my view. Are the samples metallized with Ti/Pt before the SU-8 deposition? Why? I would expect metallization to proceed after the deposition and development of the resist. Is that to test metal-polymer adhesion treatments? Is the top surface of the solar cells also metallized before photolithography? I encourage the authors to clarify these questions."

To clarify this point and give some context to our work we briefly explain our solar cell fabrication protocol. The SU-8 removal procedure here described was found accidentally during development of the following solar cell fabrication sequence: After semiconductor layer deposition in a molecular beam epitaxy reactor, the samples are metallized with 60 nm of Ti and then 5 nm of Pt by physical vapour deposition in vacuum before air exposure to avoid contamination and oxidation of the metal semiconductor interface. This metal layers serve as a seed layer for subsequent silver electroplating and improve contact resistance by avoiding contamination of the metal/semiconductor interface. These metal layers also serve as a seed layer for subsequent silver electroplating. Silicon (001) substrates were also used in experiments to test the removal speed for different SU-8 formulations. Both types of substrates were metallized with a 60 nm Ti adhesion layer and a 5 nm Pt top layer. To improve adhesion of the SU-8 photoresist to the metallized substrate, the samples were treated by immersion in a solution of 57.7 mg of 16-Mercaptohexadecanoic acid in 385 ml of ethanol, with 10 ml of deionized water, and 5 ml of acetic acid at 70 C for 10 min. Ultrasonication at 60 C for 5 min was repeated before each use of the solution to avoid the presence of agglomerates. Without this treatment partial delamination was often observed between the SU-8 photoresist and the metal surface after development of the lithography pattern. This hydrophilyc alkyl-thiol was chosen due to the well known tendency of alkyl thiols to form self assembled monolayers on noble metal surfaces. Prior to SU-8 spin coating, the samples were rinsed by spraying with isopropyl alcohol while spinning at 4000 rpm and then blow dried with nitrogen. The photovoltaic devices additionally had a pattern of silver wires electrodeposited through the lithographically defined SU-8 mask. A silver cyanide based electrolyte was used for electroplating. In our tests this electrolyte was found to be incompatible with a number of photoresists, but no effect was apparent on SU-8. Removal of SU-8 from samples not exposed to the electrolyte was also successful and no significant effect was found in the removal speed regardless of exposure to the electrolyte. After SU-8 removal, the Ti seed layer is removed with quick dip in diluted HF acid and the GaAs contact layer is removed with citric acid and hydrogen peroxide etchant.

"Section 2.2 (Polymer-metal adhesion treatment) looks a bit speculative... as it is not the main issue addressed in the paper and not even discussed in the result section, the authors could further investigate and develop this part and leave it for another publication without detracting too much from the relevance of this paper."

As suggested by the referee, we have withdrawn this section of our manuscript.

"As the authors explain in section 3.2, thermodynamical data from their DFT calculations does not proof that the proposed mechanism is the actual mechanism behind the observed results as transition states may play an important role and favor other reaction pathways. To my view and because of the technological relevance of the developed process, it is enough for the present work to prove that this is a feasible and likely mechanism with the performed calculations. However, the quality and significance of this work would be greatly enhanced if the authors provide further evidence supporting their hypothesis, either by more sophisticated simulations or by characterization of the dissolution with analytical chemistry methods."

We have found the characterization of the reaction products challenging because of these being diluted in concentrated sulfuric acid. Our attempts at neutralizing the acid have resulted in an insoluble black deposit. Characterizing the products without changing the pH seems more promising as it avoids the risk of altering the reaction products, but these restricts the suitable analytical techniques to non contact methods such as NMR. This strategy presents other difficulties, as the fact that the low pH broadens NMR peaks (see for example Batamack, P., Fraissard, J. Proton NMR studies on concentrated aqueous sulfuric acid solutions and Nafion-H. Catalysis Letters 49, 129–136 (1997). https://doi.org/10.1023/A:1019077910277).

Our assessment is that studying the reaction kinetics in the present case by calculating the energy barriers associated to all possible transition states would be computationally very costly and well beyond our capabilities. The approach by Sturgeon et al. (ref. 22) in studying ether cleavage in a similar system was to use semi-empirical methods to identify all possible stable reaction intermediates and then use the same DFT methodology we have used to calculate the energies of the most relevant intermediates. This does not give information on kinetics but illuminates the plausibility of different reaction pathways. Semi-empirical methods cannot be used reliably to identify transition states, and this makes the exploratory phase of the study overly costly. The whole exercise becomes pointless if it cannot be proved that all possible transition states have been considered. This is doable with a reasonable cost in simpler cases, but perhaps not in the present case.

We are grateful for the effort made by the reviewers and the useful feedback.

---

## Round 2 · List of Changes

The most relevant chages are:

  • Electro microscopy (SEM) images before and after SU-8 removal are now included.

  • The sub-section "Polymer-metal adhesion treatment" and all related claims have been withdrawn from the paper, although full details of the sample preparation procedure are still given in the "Sample preparation" sub-section. The abstract has also been modified accordingly.

  • Additional details about our solar cell research are now given in the "Sample preparation" sub-section to put the present results in context and clarify some of the questions asked by the reviewers.

  • We have included a discussion of the interdependence of the various experimental factors during SU-8 exposure and removal.

  • Scale bars are including in all microscopy images.

  • New references have been included as suggested, in particular in regard to SU-8 chemistry and reaction of amines with epoxides.

---

## Editorial Decision

published